# Study on Electrical Characteristics and ECG Signal Acquisition Performance of Fabric Electrodes Based on Organizational Structure and Wearing Pressure

**DOI:** 10.3390/mi16070821

**Published:** 2025-07-17

**Authors:** Ming Wang, Jinli Zhou, Ge Zhang

**Affiliations:** 1School of Chemical and Printing-Dyeing Engineering, Henan University of Engineering, Zhengzhou 450007, China; jerry_wm@haue.edu.cn; 2College of Intelligent Textile and Fabric Electronics, Zhongyuan University of Technology, Zhengzhou 450007, China; 6684@zut.edu.cn

**Keywords:** wearing pressure, fabric tissue structure, fabric electrode, ECG signal

## Abstract

Obtaining stable ECG signals under both static and dynamic conditions, while ensuring comfortable wear, is a prerequisite for fabric-electrode applications. It is necessary to study the wearing pressure of fabric electrodes as well as their organizational structure. In this study, fabric electrodes with different organizational structures (plain weave, twill weave, and satin weave) were prepared using silver-plated nylon conductive yarns as weft yarns and polyester yarns as warp yarns. The electrical characteristics of these structures of fabric electrodes were analyzed under different wearing pressures (2 kPa, 3 kPa, 4 kPa, and 5 kPa), and their effects on the quality of static and dynamic ECG signals acquired from human body were examined. The results showed that the contact impedance of the twill and satin weave structured electrodes with the skin was smaller and more stable than that of the plain weave structured electrodes. Furthermore, when a wearing pressure of 3–4 kPa was applied to the satin-structured electrodes, they not only provided satisfactory comfort but also collected stable static and dynamic ECG signals during daily exercise. These results can provide a reference for the application of fabric electrodes in ECG monitoring devices and an important basis for the design of intelligent ECG clothing.

## 1. Introduction

In recent years, cardiovascular and cerebrovascular diseases have become increasingly prevalent among younger populations. Simultaneously, the elderly population is expanding at an accelerated rate [1]. By monitoring the heart rate of patients with heart disease in real-time to detect cardiac abnormalities and alert them to take appropriate treatment measures, the mortality rate of heart disease can be greatly reduced [2].

At present, cardiac monitoring equipment used in hospitals, such as cardiac monitors and Holter machines, cannot meet the needs of most cardiac patients for long-term, real-time, daily cardiac monitoring. For example, patients with heart disease need to be monitored in a fixed place in real time when using traditional cardiac monitoring equipment for long-term cardiac monitoring, and their daily behavioral activities are greatly restricted [3]. At the same time, due to prolonged cardiac monitoring, patients with heart disease may feel itchy skin and different degrees of inflammation on the skin surface [4].

In recent years, mobile medical devices based on textile-structured, flexible ECG electrodes have emerged as potential solutions. Wearable fabric electrodes serve as flexible medical sensors developed using textile technology that enable the sensing of biological electrical signals on the human body’s surface [3,4,5]. Fabric electrodes primarily consist of conductive materials intertwined with traditional textiles or conductive materials adhered to yarns or fabrics through various coating processes to form diverse fabric electrode structures [6,7,8,9].

The reliability of ECG diagnostics heavily depends on the electrode performance. Key parameters for evaluating ECG electrodes include (As shown in Table 1): (1) skin-electrode impedance (affecting signal fidelity), (2) motion artifact resistance (critical during patient movement), (3) long-term stability (for prolonged monitoring), and (4) skin compatibility (reducing irritation risks). Optimizing these factors ensures accurate signal acquisition in clinical and ambulatory settings.

Wearable fabric electrodes, which require direct skin contact, yield varying results due to their differing tissue structures and induced contact pressure. Such differences can affect not only the effective skin and electrode contact area, thereby influencing the quality of the ECG signal, but may also impact wearer comfort [12,21]. Beckmann et al. [19] explored the ramifications of differing tissue structures on fabric electrode performance and discovered that contact impedance was significantly impacted by tissue structure, with knitted structures demonstrating lower impedance compared to others. Kusche R [10] explored various dry electrodes, including gold, stainless steel, carbon rubber, and metallized textile as contact materials, and discovered that the latter yielded the highest impedance. In some cases, however, the comfort of textile electrodes takes precedence. Similarly, research conducted by Dong Ke et al. [13] consisted of studying four types of fabric electrodes, designed with a silver-plated nylon and polyester blend interwoven plain and satin structures. Their findings suggested that a higher wearing pressure resulted in ECG signals of higher quality, while increases in non-conductive yarn content when the wearing pressure was low resulted in diminished ECG signal quality. Additionally, the quality of ECG signals collected by the fabric electrodes improved when the wearing pressure was increased. Fabric electrodes can be integrated into clothing for direct contact with skin during ECG signal measurement, which requires consideration of wearer comfort within a pressure range of 1.96–3.92 kPa [16,17,18]. Exceeding this range can cause discomfort [13]. Taji B [11] examined the impact of pressure on skin-electrode impedance and found it to be more pronounced in dry than in wet electrodes, an advantage that diminishes over extended monitoring periods. Chansaengsri K [14] demonstrated that their fabricated electrodes function without undue pressure or an electrolyte gel layer, facilitating the development of portable devices. However, optimizing this critical pressure balance requires advanced quantification methods. Recent developments in textile-based pressure sensors, which directly address the electrode-skin mechanical interface, provide essential measurement capabilities and design insights relevant to ECG electrodes. For instance, Sundarsingh EF [22] developed Zinc Oxide (ZnO) nanostructures on fabric substrates through a hydrothermal technique for use as textile pressure sensors. Kim Y [23] introduced highly conductive, flexible electrodes made from electrochemically exfoliated graphene (EEG) on cotton textiles, showing potential for piezoresistive-type pressure sensors. These methodologies are particularly valuable because they enable real-time pressure monitoring within the same textile platform as ECG electrodes, addressing the mechanical interaction challenges noted in studies like Xu [9] and Dong [13]. Therefore, understanding the pressure exerted by the fabric electrode on ECG signal quality is crucial to optimize wearable health monitoring devices and enhance the design and reliability of fabric electrodes.

Textile electrodes, serving as the crucial interface between humans and electronic devices, have shown remarkable promise in health monitoring, telemedicine, and motion tracking applications. Examples of research in this field include Nigusse AB [15], who provided a comprehensive overview of textile electrodes and their evolutionary trajectory in ECG monitoring. WU [24] focused on a silver-coated cotton/nylon fiber textile electrode suitable for ECG in healthcare systems. However, existing products often sacrifice wearer comfort for signal accuracy, especially during extended use or vigorous movement. Furthermore, while many electrodes function adequately in static situations, their performance is greatly compromised in dynamic settings, such as while walking, running, or engaging in daily activities, which hampers their practicality.

This research endeavors to address these limitations by innovating textile electrodes with various structures to significantly improve user comfort without compromising the reliability of electrocardiographic (ECG) signal monitoring in both stationary and active conditions. To this end, the study employs silver-plated nylon conductive yarn interwoven with polyester yarn using different weaves (plain, twill, and satin) to create electrodes that cater to both comfort and performance demands. The investigation involves measuring skin-electrode contact impedance under distinct pressures (2 kPa, 3 kPa, 4 kPa, and 5 kPa) and assessing ECG signal quality in both static and dynamic states to examine how wearing pressure affects the performance of electrodes with varied textile configurations.

## 2. Experiment

### 2.1. Materials and Instruments

Materials: Silver-plated nylon filament (70D, 3.8 Ω/cm) provided by Shanghai Pusheng Textiles limited company (Shanghai, China), polyester yarn (40s) purchased from Datang chemical fiber factory (Shaoxing, China), physiotherapy-grade elastic bandage provided by Sanji Medical device Technology Company (Dongguan, China), disposable gel electrode purchased form Minnesota Mining and Manufacturing (Saint Paul, MN, USA), and conductive paste provided by Weaver Medical Technology Company, Knoxville, TN, USA.

### 2.2. Design and Manufacture of Fabric Electrode Organization Structure

The weaving process was used to prepare electrode fabrics. Plain, twill, and satin weaves were produced on an SGA598 semi-automatic weaving machine (Jiangyin Tongyuan Textile Machinery Co., Ltd., Shaoxing, China), utilizing 70D silver-plated nylon filament as weft and polyester yarn (40s) as warp. Post-weaving, electrodes were cut into 3 cm diameter circles using a precision die-cutting tool. This diameter aligns with standard ECG electrode sizes used in clinical practice [24] and prior textile electrode studies [9,13], optimizing the balance between:(1)Adequate skin contact area for stable signal acquisition [9,11].(2)Wearability and comfort during dynamic activities [13,17].(3)Mitigation of motion artifacts through controlled interfacial stability [9,23].

Edge sealing was applied to prevent yarn fraying. The preparation process is shown in Figure 1. Fabric weave diagrams and physical photographs are shown in Figure 2.

### 2.3. Testing and Characterization

#### 2.3.1. Surface Resistance Test Between Conductive Fabric and Copper Plate

A VICTOR 86E digital multimeter (Xi’an Beicheng Electronics Co., Ltd., Xi’an, China) was used to test the frequency sweep impedance of a single electrode and a copper plate to evaluate the surface conductivity of the fabric electrode. The frequency range of the test was 0.1~1000 Hz, and the voltage was 5 mV. Each fabric sample was measured five times, and the average value was taken as the test result.

Since the conductive yarns inside the fabric-type electrodes mostly present a three-dimensional buckling state, applying different pressures to the electrodes will cause different degrees of changes in the internal structure of the electrodes, thereby resulting in different electrical properties of the electrodes. Therefore, when conducting surface resistance tests on the electrodes, it is necessary to ensure that the pressure conditions are consistent for all samples being tested. Connect one end of the multimeter to the left side of the copper plate, and the other end to the electrode transmission line (the resistance of the transmission line can be ignored). Place the fabric electrode against a clean copper plate, then place an insulated weight of 1 kg on the back of the embroidered fabric electrode to completely cover the electrode and make it adhere tightly to the copper plate, as shown in Figure 3. Note that the tested electrode should be placed at the center of the insulated weight to ensure uniform force on the electrode and reduce experimental errors. Set the test frequency to 0.1–1000.0 Hz and the test voltage to 5 mV.

#### 2.3.2. Impedance Test of Contact Between Fabric Electrode and Skin

The IM3533-01 LCR tester (HIOKI Company of Japan, Nagano, Japan) was utilized to measure the contact impedance between the skin and electrode under varying pressures. During the test, two fixed-distance points (90 mm apart) were marked on the arm, and two fabric electrodes with the same organizational structure were positioned at these points. Subsequently, the AMI airbag contact pressure gauge’s two airbags were placed adjacent to each electrode and secured using an elastic band. The contact pressure between the skin and electrodes is adjusted via this elastic band, enabling measurement of the contact impedance across different tissue structures at 2 kPa, 3 kPa, 4 kPa, and 5 kPa, and the voltage of the tester was set to 50 mV. A thin layer (~0.1 mm in thickness) of physiologically compatible conductive paste (Ten20™, Weaver, Aurora, CO, USA) was uniformly applied to the skin-contact surface of the fabric electrodes using a sterile spatula to improve electrode-skin interface conductivity and stabilize impedance readings. This standardized interface preparation minimizes the variable contact resistance inherent to dry textile electrodes [10,11], which is particularly crucial for controlled pressure-impedance characterization.

#### 2.3.3. Collection of Static and Dynamic ECG

The BIOPAC MP160 multi-conduction physiological recorder (16 channels, BIOPAC, Goleta, CA, USA) was utilized to measure the ECG of the human body under both static and dynamic conditions. Before measurement, two fabric electrodes, prepared with the same organizational structure, were fixed at the left and right thoracic cavities with elastic bands and connected to the physiological recorder. The airbag of the AMI airbag contact manometer was also fixed at the position next to the fabric electrodes through the elastic band to ensure that the pressures of the fabric electrodes and the airbag remained consistent. During measurement, the electrode-skin contact pressure was adjusted by adjusting the position of the elastic band Velcro. At the same time, two disposable ECG electrodes were attached to the lower chest position, underneath the two fabric electrodes, which were also connected to the physiological recorder, and the measured ECG was compared and analyzed with the quality of the ECG measured by the fabric electrodes. The static and dynamic ECGs were measured at contact pressures of 2 kPa, 3 kPa, 4 kPa, and 5 kPa for the fabric electrodes of different tissue structures. Measuring the ECG under dynamic conditions, the electrode-skin interactions will occur, which are prone to causing motion artifacts. Pengjun Xu [9] conducted experiments on nine types of motion when studying the effect of motion on the ECG: three types of daily life motion: walking, jogging, and running; three types of trunk movement: bending, twisting, and sitting; and limb movement: forward and backward swinging of the arms, flat lifting of the arms, and expansion of the chest movement. Referring to the experiment by Pengjun Xu and combining with the experimental conditions, three types of actions were selected for this experiment: sitting down and standing up, chest expansion movement, and raising the right arm to measure the dynamic electrocardiogram.

## 3. Results and Discussion

### 3.1. Analysis of Surface Resistance Between Conductive Fabric and Copper Plate

The results of the surface resistance test between the conductive fabrics with different organizational structures and copper plates are shown in Figure 4. It was found that the minimum surface resistance was formed between the plain structure conductive fabric and the copper plate. Satin and twill structures have longer floating threads than plain structures, while yarns flex more in plain organization. The capacitance of the double layer formed by the yarn in the highly flexed state was larger than that formed by the yarn in the straight state. Therefore, the surface resistance between the plain organization structure conductive fabric and the copper plate was the smallest.

### 3.2. Analysis of the Impedance of the Contact Between the Fabric Electrode and the Skin

The structural composition of the fabric notably influences the contact impedance when interfacing with the skin as an electrode. Plain weave, twill weave, and satin weave represent prevalent textile constructions, each distinguished by its unique method of interlacing yarns. These differential weaving techniques critically define the resultant physical attributes of the fabrics, such as elasticity, breathability, and thickness [25]. When deployed as electrodes in applications such as biomedical sensors, these inherent material properties significantly affect the level of contact impedance with the skin.

In this experiment, the fabric electrodes with three organizational structures (plain weave, twill weave, and satin weave) were used. The wearing pressure range was set at 2, 3, 4, and 5 kPa for testing contact impedance between the electrode and skin under different tissue structures or pressures and was compared with that of disposable gel electrodes.

#### The Influence of Tissue Structure on the Impedance of the Contact Between the Fabric Electrode and the Skin

The structure of the fabric critically influences the electrocardiogram (ECG) signal obtained from fabric electrodes [20]. Under identical wearing pressures (e.g., 2, 3, 4, and 5 kPa), fabric electrodes with varied weaves manifest differing contact impedance profiles with skin across frequencies, as depicted in Figure 5. Analysis of Figure 5a–d highlights a trend of diminishing impedance values with increasing frequency across all fabric electrodes, with satin and twill weaves showing lower and more consistent impedance than their plain counterparts. Additionally, the inclusion of non-conductive yarn leads to a marked elevation in impedance. Notably, at lower frequencies (0.5–100 Hz), the skin contact impedance of satin and twill fabric electrodes outperforms that of disposable gel electrodes, showcasing their stability.

### 3.3. Quality Analysis of Static and Dynamic ECG

In practical applications, optimizing fabric electrode performance often entails adjusting wearing pressure to maintain effective skin contact. Contact impedance serves as a crucial metric for assessing electrode efficacy, particularly in medical and sports scenarios that require long-term physiological signal monitoring. Consequently, this study investigates fabric electrodes with diverse structural compositions, records electrocardiograms in both static and dynamic states, and subsequently analyzes signal quality to elucidate performance disparities stemming from varying tissue structures.

#### 3.3.1. Analysis of the Quality Results of Static ECG Collected by Fabric Electrodes

The impact of fabric electrodes with varying weave structures, including plain, twill, and satin, on static ECG quality is predominantly manifested in signal stability, noise levels, and signal clarity. These variables are directly related to the contact quality between the fabric electrode and the skin, with the fabric’s structural design being the principal determinant of this interaction.

During the static lower ECG acquisition, the tester remained stationary, and the fabric electrodes on the chest remained stable. Figure 6 shows that the static lower ECG acquired by the fabric electrodes and disposable gel electrodes, with red and green waveforms, respectively, representing each electrode (ECG traces from the textile electrode are shown in red; traces from the commercial soft gel electrode are shown in green). The results indicated that the fabric electrodes can produce high-quality ECG signals regardless of the organizational structure or wearing pressure and can accurately identify the P-waves, T-waves, and QRS wave groups, which were consistent with the ECG signals obtained from disposable gel electrodes in the resting state.

Figure 6 reveals that with varying pressures during electrocardiography, the plain fabric electrode’s P-wave is blunt and circular, maintaining an amplitude below 0.25 mV. This observation, mirroring the gel electrode’s P-wave, confirms the fabric’s accurate capture of atrial depolarization. Although the complete QRS complex is distinctly discernible, the R-wave amplitude shows a slight, gradual decline as pressure increases, suggesting potential influences on the subject’s autonomic nervous system balance or electrode displacement. These changes raise the possibility of sympathetic activation or parasympathetic inhibition, or mechanical alterations, such as electrode movement, which could compromise the accuracy of R-wave amplitude measurements. Therefore, the progressive decrease in R-wave amplitude with escalating pressures could result from multiple sources. This complex interplay underscores the importance of accounting for physiological and pathological states, technical aspects, and individual patient differences when analyzing such phenomena.

Due to its tight and orderly weave, the plain weave structure offers an ample contact area and low impedance under static conditions, which aids in diminishing contact noise. As a result, the ECG signal is rendered more stable and distinct, rendering plain weave fabric electrodes ideal for monitoring scenarios that necessitate elevated signal fidelity, particularly in static contexts.

Figure 7 depicts the electrocardiogram obtained from a twill fabric electrode under varying pressures, where the P-wave appears blunt and circular with an amplitude under 0.25 mV, akin to the response from a plain fabric electrode. This similarity to the gel electrode’s reading suggests an accurate depiction of atrial depolarization. Notably, the QRS complex remains clear, yet the R-wave amplitude exhibits erratic changes with increasing pressure. These variations may stem from electrode movement or instability during testing, underscoring the fabric electrodes’ potential application in wearable sensors. Moreover, in comparison to the plain fabric electrode, the twill electrode shows a marginally higher peak R-wave amplitude.

The twill weave offers superior elasticity and adaptability, enhancing the fit over various body contours. Such conformability minimizes air gaps, even under static conditions, thereby attenuating signal distortion associated with movement or positional shifts. Consequently, twill fabrics tend to exhibit improved anti-interference capabilities in ECG monitoring applications, particularly during minor patient movements.

Figure 8 demonstrates that the cardiac signal captured by the satin fabric electrode closely parallels that obtained using a gel electrode. The P-wave retains the blunt and circular shape noted in other fabric electrodes, with an amplitude of less than 0.25 mV. Nevertheless, the peak R-wave amplitude is somewhat reduced in comparison to that recorded by both plain and twill fabric electrodes.

The satin fabric’s smooth surface and fewer interlacing points tend to diminish the microscopic contact points with the skin in a static state, potentially resulting in marginally elevated contact impedance. Increased contact impedance can amplify background noise, adversely impacting ECG signal quality. Nonetheless, the high smoothness of satin fabric may facilitate even pressure distribution, occasionally stabilizing the signal. During the selection of the fabric electrodes for ECG monitoring, the emphasis should be placed on the fabric’s physical attributes and its interaction with the skin. Typically, plain weave fabrics offer stable and distinct signals, whereas twill fabrics may excel in dynamic scenarios, and satin fabrics may necessitate specialized design considerations to enhance contact efficacy.

#### 3.3.2. Quality Analysis of Dynamic ECG Collected by Fabric Electrodes Under Different Wear Pressures

The fabric structure of electrodes considerably affects the quality of electrocardiograms (ECGs), particularly under dynamic conditions such as patient movement or physical activity. Plain, twill, and satin weaves each exhibit distinct advantages and limitations in these scenarios.

Figure 9, Figure 10 and Figure 11 show the dynamic lower ECG collected by the fabric electrodes. In this case, the red waveform was the ECG collected by the fabric electrode, and the green waveform was the ECG collected by the disposable gel electrode. Figure 9, Figure 10 and Figure 11 show that the motion of the lower limbs while sitting and standing had little effect on the quality of the collected ECG signals. Even at a low wear pressure, the QRS waves could still be recognized by the ECG. This was due to the fact that the fabric electrodes were in contact with the chest, while the movement of the lower extremities caused them to slip slightly. At the same time, the quality of the ECG collected by a satin electrode was better than that of other tissue-structured electrodes, with clearer and more stable waveform maps.

The expansion of the chest and the raising of the right arm belonged to the limb movements. The test results are shown in Figure 9b,c, Figure 10b,c, and Figure 11b,c, respectively. These two movements had a significant effect on the quality of the ECG collected under the wearing pressure of 2 kPa and 3 kPa. When the right arm was moved upward, the P-wave, T-wave, and QRS waves were not recognized in the ECGs acquired with the plain structure electrodes at 2 kPa and 3 kPa. On the other hand, at wearing pressures of 4 kPa and 5 kPa, ECGs acquired with electrodes of different tissue structures for three motions were basically able to clearly distinguish QRS waves, except for P- and T-waves, which could be recognized by the plain electrodes. Therefore, it can be concluded that the quality of ECG signals acquired at larger wear pressure was superior to that of ECG signals acquired at smaller pressure. According to the pressure range of the human body comfort clothing (1.96–3.92 kPa), the optimal pressure range is 3–4 kPa.

Figure 9, Figure 10 and Figure 11 compare the electrocardiogram data obtained from a fabric electrode under variable dynamic conditions to that from a disposable gel electrode (indicated by a green waveform chart). These images effectively demonstrate the fabric electrodes’ efficacy in ECG monitoring, particularly their stability amid physical activity.

Figure 9a illustrates an electrocardiogram recorded using a plain fabric electrode during a seated exercise. The data show that even under low wearing pressure, the electrode can accurately capture the QRS complex, maintaining firm contact with the chest and minimizing slippage due to lower limb movements.

Subsequent Figure 7c and Figure 9b indicate that movements such as chest expansion and right arm elevation are more susceptible to changes in wearing pressure. At pressures of 2 kPa to 3 kPa, there is a notable degradation in signal quality with the plain fabric electrode, rendering identification of the P, T, and QRS components challenging, which underscores the critical role of wearing pressure in securing reliable ECG readings. Conversely, at increased pressures ranging from 4 kPa to 5 kPa, signal recognition deteriorates even further, with the QRS complex, P waves, and T waves becoming less distinctive.

The structure of plain weave fabric is robust; however, its limited elasticity can lead to inadequate electrode-skin contact during vigorous exercise, potentially resulting in intermittent signals. Nevertheless, when fitted correctly, plain weave fabric is capable of sustaining good contact during activities of low to moderate intensity, thus preserving signal stability.

Comparing Figure 9 and Figure 10, the twill fabric electrode exhibits performance akin to the plain fabric electrode for seated exercises. However, the ECG quality at 2 kPa to 3 kPa pressure settings is enhanced with the twill structure, outperforming the plain electrode and underscoring the significance of the textile structure. At elevated pressures of 4 kPa to 5 kPa, the twill electrode excels in delineating the QRS complex and identifying P and T waves across all tested movements, attesting to the benefits of appropriately adjusted wearing pressures in enhancing signal quality.

Twill fabrics excel in dynamic conditions owing to their enhanced elasticity and adaptability, which allow for superior accommodation of muscle movement and alterations in body surface topology. Consequently, this reduces movement-induced signal distortion. Furthermore, twill’s structure inherently diminishes friction between the fabric and the skin, thereby mitigating the occurrence of false signals attributable to friction.

Relative to the previous two fabric electrodes, the satin structure electrode yields a more distinct and stable electrocardiogram waveform, demonstrating its enhanced performance in dynamic monitoring during various movements, such as sitting up, chest expansion, and raising the right arm.

While satin fabric’s smoothness may slightly elevate contact impedance in static conditions, it also lowers friction and irritation during dynamic activities, enhancing comfort. Yet, its stability may falter compared to plain or twill weaves in high-intensity sports, necessitating specialized designs—such as advanced fixing methods or material combinations—to bolster its fit and steadiness. Upon considering all aspects, twill fabric electrodes outperform others in dynamic scenarios, whereby high flexibility and minimal motion artifacts are critical.

#### 3.3.3. Quality Analysis of Dynamic ECG Collected by Electrodes of Different Tissue Structures Under the Same Wearing Pressure

Dynamic electrocardiograms obtained using electrodes of varying tissue structures, under uniform wearing pressures (2 kPa, 3 kPa, 4 kPa, and 5 kPa), are depicted in Figure 12, Figure 13, Figure 14 and Figure 15.

Figure 12 demonstrates that at lower pressures, the electrocardiogram quality from satin structure electrodes surpasses that of twill and plain weaves, with plain weave exhibiting the poorest results. Additionally, ECGs recorded while the right arm moves upward, under pressures of 2 kPa, 3 kPa, and 4 kPa, do not distinctly reveal the QRS complex. This ambiguity can be attributed to plain electrodes typically yielding more stable ECG signals due to their uniform skin contact, which is beneficial for prolonged monitoring. However, excessively dense fabric could cause signal distortion. Twill electrodes may provide a preferable fit and consequently enhance signal fidelity due to their structure; yet, their inherent elasticity could result in fluctuating contact areas under sustained load, potentially destabilizing the signal. Conversely, satin’s smooth surface might decrease impedance and improve ECG quality. Nevertheless, its looser, more unstable weave may become deformed, snagged, or worn after extended use and pressure, which can compromise signal continuity and precision. A comprehensive analysis indicates superior dynamic ECG signal quality from electrodes with a satin weave structure.

Electrocardiographic testing at 3 kPa exhibits similarities to results at 2 kPa. Ideally, the plain electrode would generate the most stable ECG signal, suitable for extended monitoring. Yet, within the low-to-medium pressure parameters of this study, a high fabric density compromised the signal integrity, leading to distortions. The twill electrode’s structural design potentially offers a snugger fit and superior signal quality, though its elasticity also presents a paradox; while facilitating closer skin contact, it may induce fluctuations in the electrode-skin interface, thereby impairing signal stability.

In contrast, the satin electrode’s smooth surface significantly lowers skin-to-electrode impedance, reliably transmitting high-quality ECG signals. Despite its looser structure, which could pose risks of deformation or material wear over extensive usage or under substantial pressure, the electrode maintained optimal signal continuity, clarity, and precision within the experimental pressure range, substantiating the satin configuration’s superiority in Holter electrocardiography.

At an increased pressure of 4 kPa, the twill electrode effectively harnesses its elasticity and durability, accommodating minute dermal variations and preserving a consistent, snug contact. This attribute notably contributes to stable signal transduction, mitigating interference from potentially uneven pressure.

At an applied pressure of 5 kPa, the electrocardiogram (ECG) quality obtained from electrodes with the three textile structures—plain, twill, and satin—is satisfactory. The subsequent discussion analyzes the dynamic ECG quality from these electrodes under this increased pressure. The plain weave electrode boasts a stable architecture; however, its comparatively limited elasticity may lead to non-uniform skin contact at higher pressures, engendering potential signal distortion or amplification of noise. Nevertheless, with an even pressure distribution, plain weave can still yield a reasonably consistent ECG signal. Twill weave electrodes exhibit both elasticity and durability, contributing to an improved conformity to skin irregularities. The presence of higher pressures leverages twill’s elasticity to enable sustained, uniform skin contact, attenuating signal disruptions from localized pressure points. Consequently, twill may resist deformation from wear, thereby enhancing signal quality. Meanwhile, satin weave electrodes, known for their smooth texture and enhanced comfort, are disadvantageously less robust and elastic. Under significant pressures, the gentle structure and constrained elasticity can lead to deformation, diminishing the skin contact area and compromising signal fidelity and stability. Moreover, sustained high pressures may impair the electrode’s fabric, further detracting from the reliability of ECG readings.

When selecting fabric electrodes that will be subjected to considerable pressure, twill electrodes are preferable due to their superior elasticity and resilience while preserving signal integrity. Plain weave electrodes are a secondary choice, particularly for scenarios prioritizing signal steadiness. Satin-weave electrodes are most apt for applications with less stringent pressure demands and a greater emphasis on comfort. Throughout the selection process, the electrode’s material, engineered design, and interface interaction with skin should also be carefully considered.

## 4. Conclusions

In this paper, silver-plated nylon conductive yarn was selected as the weft yarn, and polyester yarn was used as the warp yarn to prepare conductive fabric electrodes with different organizational structures (plain weave, twill weave, and satin weave). The electrical properties of the fabric electrodes under four different wear pressures and the quality of the acquired human static and dynamic ECGs were compared and analyzed. The following conclusions were drawn:(a)The contact impedance between electrodes of the same organizational structure and the skin decreased with increasing wearing pressure. Satin and twill structure electrodes had lower contact resistance with the skin than plain fabric electrodes, and their impedance curves changed more smoothly.(b)The organizational structure and the wearing pressure of the fabric electrodes had little effect on the quality of the ECG under rest. Dynamical ECG signals acquired at larger wearing pressures were better than those at smaller pressures. The best ECG signals were acquired by the satin structure electrodes compared to the other two structure fabric electrodes.(c)Combined with the pressure range of human comfortable clothing, it was concluded that the fabric electrodes prepared with the satin structure can collect dynamic and static ECG signals with stable quality. The satin-structure fabric electrodes simultaneously provided good comfort wearing in the pressure range of 3–4 kPa.

## Figures and Tables

**Figure 1 micromachines-16-00821-f001:**
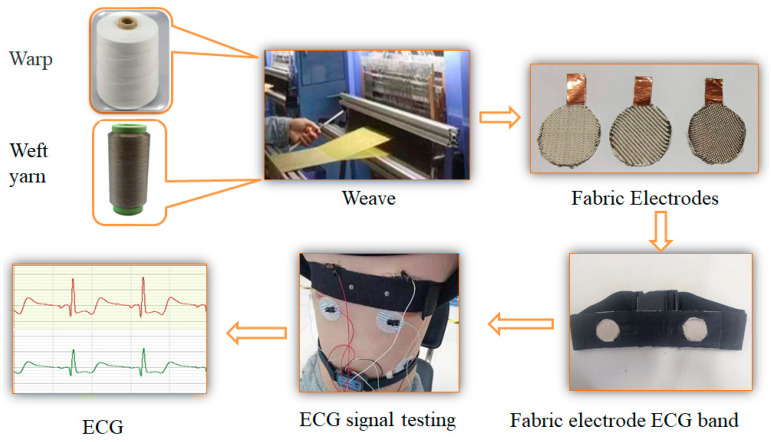
Preparation and test process diagram of the fabric electrode.

**Figure 2 micromachines-16-00821-f002:**
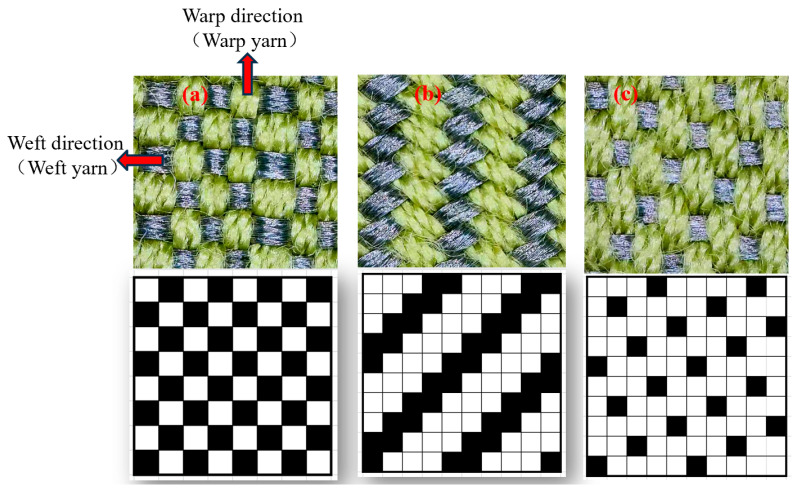
The fabric’s weave, diagram, and Physical photograph: (**a**) Plain weave; (**b**) 2/3 right twill; (**c**) 3/5 weft satin.

**Figure 3 micromachines-16-00821-f003:**
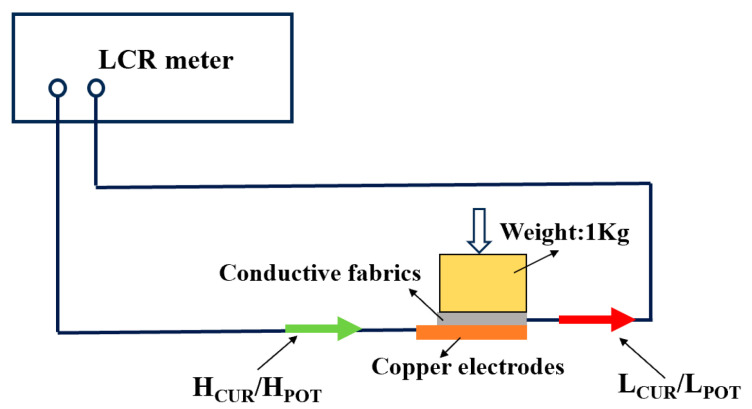
Schematic diagram of surface resistance test between conductive fabric and copper plate. (H_CUR_ is Current output terminal, H_POT_ is High side voltage detection terminal, L_CUR_ is LOW side voltage detection terminal, and L_POT_ is Current detection terminal).

**Figure 4 micromachines-16-00821-f004:**
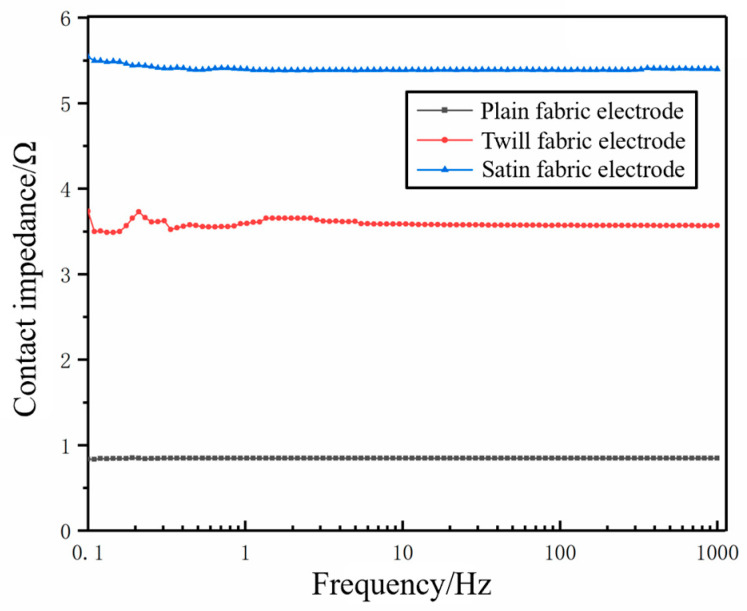
The contact impedance conductive fabric and copper plate.

**Figure 5 micromachines-16-00821-f005:**
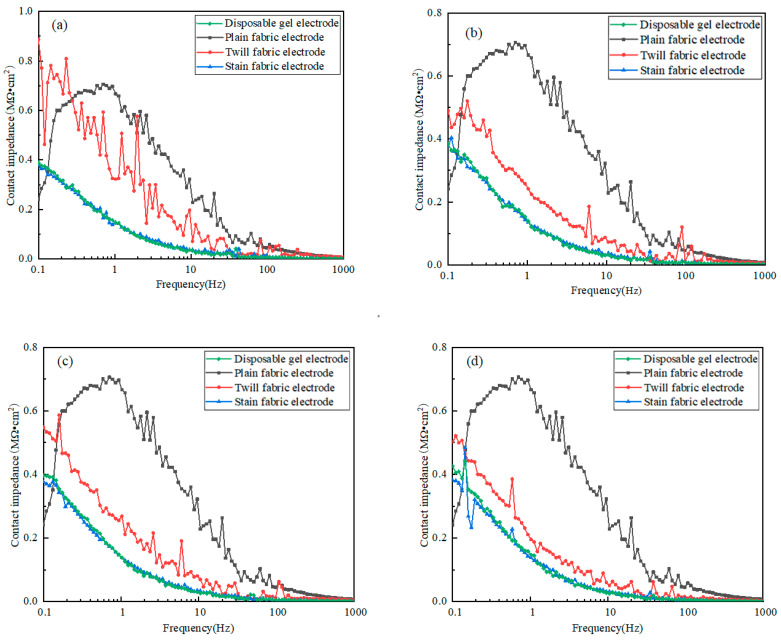
The impedance diagram of the contact between the electrode and the skin of different tissue structures under the same wearing pressure: (**a**) 2 kPa; (**b**) 3 kPa; (**c**) 4 kPa; (**d**) 5 kPa.

**Figure 6 micromachines-16-00821-f006:**
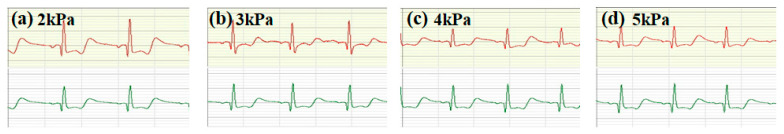
Static electrocardiogram collected by plain fabric electrode (ECG traces from the textile electrode are shown in red; traces from the commercial soft gel electrode are shown in green).

**Figure 7 micromachines-16-00821-f007:**
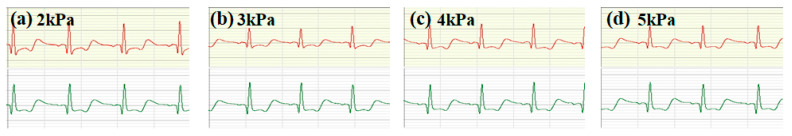
Static electrocardiogram collected by twill fabric electrode (ECG traces from the textile electrode are shown in red; traces from the commercial soft gel electrode are shown in green).

**Figure 8 micromachines-16-00821-f008:**
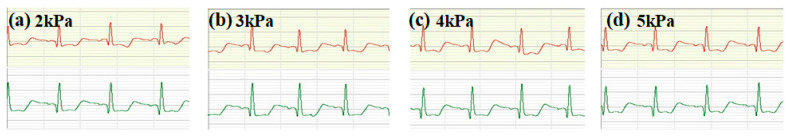
Static electrocardiogram of satin fabric electrode (ECG traces from the textile electrode are shown in red; traces from the commercial soft gel electrode are shown in green).

**Figure 9 micromachines-16-00821-f009:**
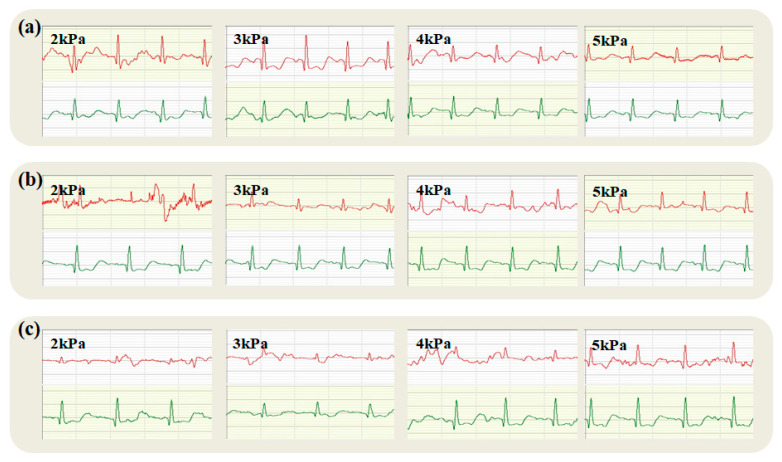
The dynamic ECG collected by the plain tissue structure fabric electrodes under different wearing pressures: (**a**–**c**) were sitting and standing, chest expansion, and right arm upward movements, respectively (ECG traces from the textile electrode are shown in red; traces from the commercial soft gel electrode are shown in green).

**Figure 10 micromachines-16-00821-f010:**
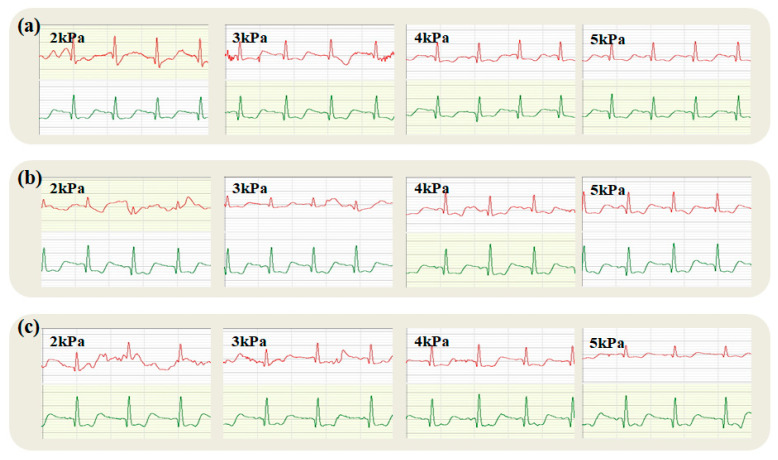
The dynamic ECG collected by the twill tissue structure fabric electrodes under different wearing pressures: (**a**–**c**) were sitting and standing, chest expansion, and right arm upward movements, respectively (ECG traces from the textile electrode are shown in red; traces from the commercial soft gel electrode are shown in green).

**Figure 11 micromachines-16-00821-f011:**
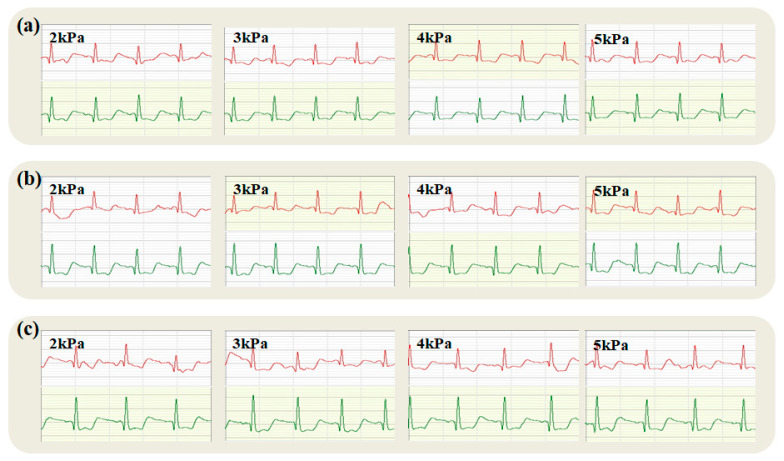
The dynamic ECG collected by the satin tissue structure fabric electrodes under different wearing pressures: (**a**–**c**) were sitting and standing, chest expansion, and right arm upward movements, respectively (ECG traces from the textile electrode are shown in red; traces from the commercial soft gel electrode are shown in green).

**Figure 12 micromachines-16-00821-f012:**
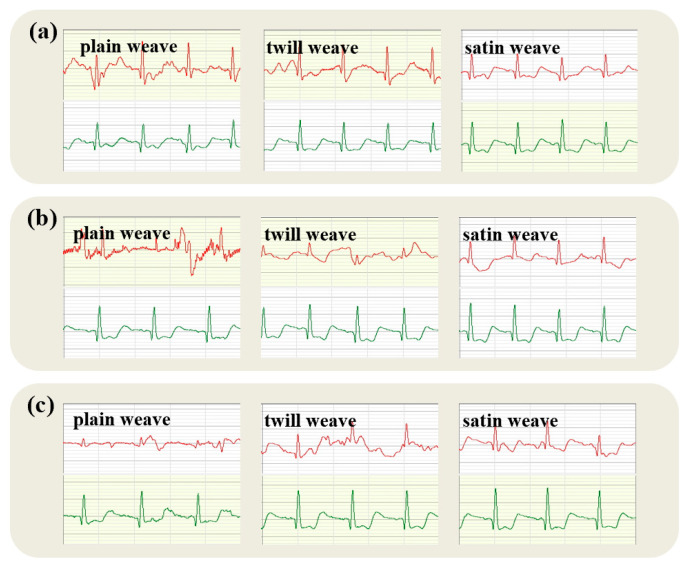
Dynamic ECGs collected by fabric electrodes with different tissue structures under 2 kPa pressure: (**a**–**c**) sitting and standing, chest expansion, and right arm upward movements, respectively (ECG traces from the textile electrode are shown in red; traces from the commercial soft gel electrode are shown in green).

**Figure 13 micromachines-16-00821-f013:**
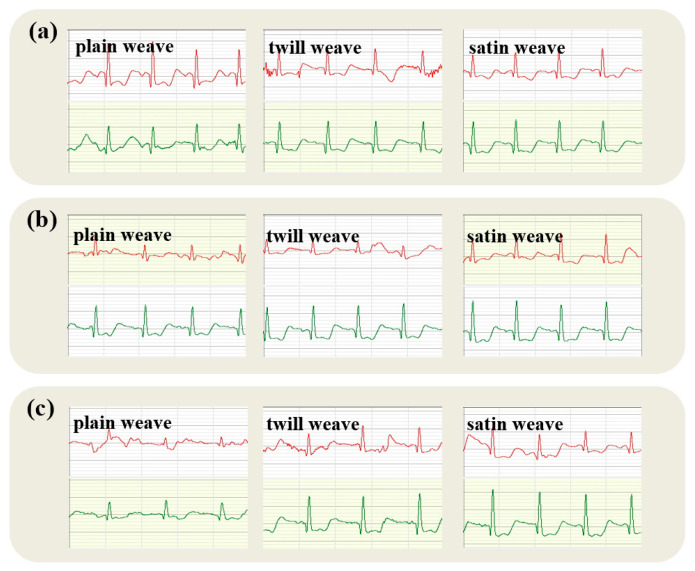
Dynamic ECGs collected by fabric electrodes with different tissue structures under 3 kPa pressure: (**a**–**c**) are sitting and standing, chest expansion, and right arm upward movements, respectively (ECG traces from the textile electrode are shown in red; traces from the commercial soft gel electrode are shown in green).

**Figure 14 micromachines-16-00821-f014:**
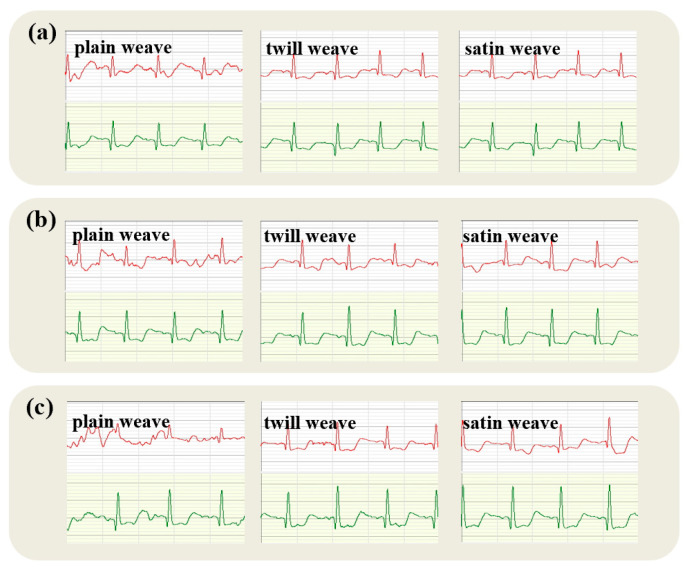
Dynamic ECGs collected by fabric electrodes with different tissue structures under 4 kPa pressure: (**a**–**c**) are sitting and standing, chest expansion, and right arm upward movements, respectively (ECG traces from the textile electrode are shown in red; traces from the commercial soft gel electrode are shown in green).

**Figure 15 micromachines-16-00821-f015:**
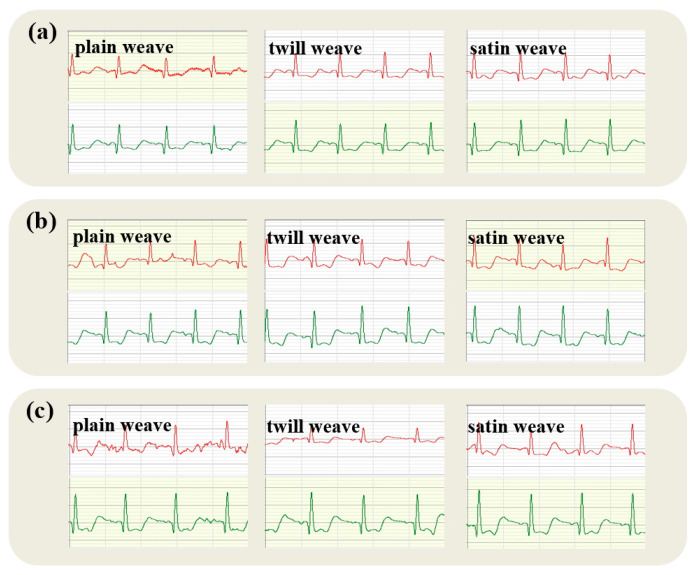
Dynamic ECGs collected by fabric electrodes with different tissue structures under 5 kPa pressure: (**a**–**c**) are sitting and standing, chest expansion, and right arm upward movements, respectively (ECG traces from the textile electrode are shown in red; traces from the commercial soft gel electrode are shown in green).

**Table 1 micromachines-16-00821-t001:** Relative parameters.

Parameter	Significance
Skin-electrode impedance	High impedance leads to signal attenuation/noise increase (ideal value: ≤2 kΩ at 10 Hz) [10,11]
Motion artifact suppression	The ability of the electrode to resist motion interference (critical for wearable ECG) [11,12,13]
Long-term stability	The consistency of signal quality over time (especially important for long-term monitoring) [14,15]
Biocompatibility	Reducing skin irritation/allergic reactions (ISO 10993 standard) [16,17,18]
Baseline drift suppression	Maintaining a stable signal baseline (affected by electrolyte gel/adhesive) [9,13]
Signal-to-noise ratio (SNR)	The intensity ratio of the pure signal to noise determines the distinguishability of the waveform. [19,20]

## Data Availability

The raw/processed data required to reproduce these findings cannot be shared at this time, as the data also forms part of an ongoing study.

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
