# Peer review of "Study on Electrical Characteristics and ECG Signal Acquisition Performance of Fabric Electrodes Based on Organizational Structure and Wearing Pressure"

_micromachines, 2025, doi:10.3390/mi16070821_

Round 1

Reviewer 1 Report

Comments and Suggestions for Authors

This manuscript reported the study on the performance of woven conductive fabric electrodes for ECG signal acquisition. Three types of woven structures (plain weave, twill weave, and stain weave) were fabricated and compared the performance under wearing pressures during static and dynamic wearing conditions. While this study is important for developing wearable textile electrodes for biopotential recording, there are several points that need clarification.

  1. On page 2 of the introduction section, the literature review of developing textile pressure sensor is not coherent within the context of that paragraph. It is suggested to add some rational/reasoning for adding the textile pressure sensor review to that part.
  2. Please specify the resistance of the conductive silver-plated nylon filament used for clarification in the experimental section.
  3. Please add scale bars in Figure 2 for clarification.
  4. How was the woven textile electrode shaped into a circular shape? The diameter of the circular fabric electrode tested was 3 cm. Why was 3 cm chosen? Does the diameter of the fabric electrode make a difference in the performance of the ECG signal acquisition?
  5. How was the surface resistance measured? Why was the copper plate chosen and how does the copper plate form the contact with the fabric electrode? A simple schematic to show the setup could be helpful.
  6. In Table 1, what frequency of impedance was used to calculate the average surface resistance?
  7. In section 2.3.2, it says “In order to obtain better signs, the media reagents (conductive paste) were used on the surface of the fabric electrodes.” This is confusing and needs more clarification. What type of conductive paste was used? Which side of the fabric was the conductive paste applied and how was the conductive paste applied? Why was the media reagent needed?
  8. It is suggested to label all the ECG graphs to show which plot is from the fabric electrode and which plot is from the soft gel electrode, and add more details in the corresponding figure captions.
  9. In line 159, there was a typo/grammar error: “three of the movement movements…”

Reviewer 2 Report

Comments and Suggestions for Authors

In this article, “Study on Electrical Characteristics and ECG Signal Acquisition 2 Performance of Fabric Electrodes Based on Organizational 3 Structure and Wearing Pressure,” the authors have fabricated electronic textile for ECG monitoring using silver plated nylon and PET yarn as weft and warp yarns. Three configurations of weave were used, including plain, satin and twill. The authors have characterized the wearable electrodes under various static and dynamic conditions with various contact pressure to ensure wear comfort. The article is a good addition to the literature of wearable e-textiles. I would recommend its publication after the following revisions.

  1. “In recent years, cardiovascular and cerebrovascular diseases have gradually become 28 younger”.Consider rephrasing this statement. Its not clear that the authors are trying to say that the cases of cardiac arrests are becoming more common among younger generation.
  2. Please discuss briefly the main parameters to evaluate the performance of ECG electrode in the introduction.
  3. Please include in the figure 1 or 2 about the basics of warp and weft yarns so that non-textile researchers and medical Dr.’s can understand.
  4. Section heading 2.3.3 Collection of static and static ECG. I think the author meant static and dynamic. Please correct the typo.
  5. A thorough proofreading will greatly improve the quality of the manuscript.

Round 2

Reviewer 1 Report

Comments and Suggestions for Authors

All the comments were addressed.